# Characterisation of Benthic Macroinvertebrate Communities in Small Watercourses of the European Central Plains Ecoregion and the Effect of Different Environmental Factors

**DOI:** 10.3390/ani12050606

**Published:** 2022-02-28

**Authors:** Adam Brysiewicz, Przemysław Czerniejewski, Jarosław Dąbrowski, Krzysztof Formicki

**Affiliations:** 1Institute of Technology and Life Sciences—National Research Institute, Falenty, 3 Hrabska Avenue, 05-090 Raszyn, Poland; j.dabrowski@itp.edu.pl; 2Department of Commodity, Quality Assessment, Process Engineering and Human Nutrition, West Pomeranian University of Technology in Szczecin, Kazimierza Królewicza 4 Street, 71-550 Szczecin, Poland; pczerniejewski@zut.edu.pl; 3Department of Hydrobiology, Ichthyology and Biotechnology of Reproduction, West Pomeranian University of Technology in Szczecin, Kazimierza Królewicza 4 Street, 71-550 Szczecin, Poland; kformicki@zut.edu.pl

**Keywords:** macrozoobenthos, stream, biodiversity, water quality, bioindication

## Abstract

**Simple Summary:**

Macroinvertebrates are very important to aquatic ecosystems. They are food for vertebrates and their biodiversity serves as a testament of the quality of their habitats. The majority of research on macroinvertebrates simply describes the composition and population density of species living in large rivers and lakes. The aim of our study was to compare the biodiversity of macrozoobenthos assemblages and to determine the effect of physicochemical and hydrological conditions on their abundance and density in 10 small rivers in agricultural areas. Overall, 105 taxa were identified as species living in changing conditions. Oxygenation of water and nitrogen content were important factors determining the existence of macroinvertebrates in these rivers. Some groups of organisms showed sensitivity to changes in water temperature as well as to the flow and depth in rivers.

**Abstract:**

Most publications on the influence of environmental factors on macroinvertebrate communities focus on large rivers, whereas relatively few examine small watercourses in agricultural areas, which, due to their size and pressure from intensive agricultural production, are much more susceptible to the effects of unfavourable environmental conditions or anthropopressure. The aim of our study was to compare the biodiversity of macrozoobenthos assemblages and to determine the effects of physicochemical and hydrological conditions on their abundance and density in 10 small rivers in agricultural areas located in northwest (53°23′ N 15°14′ E) and central (52°11′ N 20°48′ E) Poland. In total, 105 taxa were recorded, with the majority being euryoecious. Among the assessed physicochemical parameters, oxygenation was found to affect the density and number of taxa; another important factor was the content of nitrate nitrogen. Sensitivity to changes in water temperature was observed in some macrozoobenthos taxa (especially Decapoda). Of the examined hydrological parameters, the greatest effects were exerted by speed, flow, and depth.

## 1. Introduction

Lotic habitats are of extremely high ecological value and have great significance for biodiversity protection [1]; they are also the most often exploited ecosystems on our planet [2]. At the same time, such ecosystems are sensitive to, and are affected by, numerous human activities [3,4]. The degradation of river water quality under the effect of increased agricultural, communal, and industrial pollution, which constitutes a serious threat to river integrity, is among the most important reasons for the deteriorating conditions in rivers, which may have an effect in decreasing taxonomic richness and biodiversity [2,5,6]. Ecological integrity is defined as the ability to support and maintain a balanced, integrated, and adaptable community of organisms with composition and diversity comparable to those of natural habitats within the region [7]. Changes in the water quality and physical structure of a riverbed and habitats, as a consequence of anthropogenic effects, alter the composition of the river’s biotic assemblage, usually resulting in a decrease in its species richness and diversity [8,9]. Macroinvertebrates are among the sensitive groups that react to both natural and anthropogenic changes in the river environment [10,11]. Due to their sensitivity to oxygen concentration and changes in chemical properties of the water [12], food availability [13], and changes in the habitat structure [14], they are most often used in ecological assessment as an indicator of the quality of various aquatic environments [15,16]. Assessment of water quality is more effective when applying biological rather than physicochemical methods, which describe the water quality only over short periods or as a single ‘snapshot’ of conditions [17]. By contrast, changes in zoobenthos structure reflect long-term changes in water quality [18] due to the association between physicochemical changes in the aquatic environment and the sensitivity of particular taxa [1,19,20]. Invertebrates, being sensitive to chemical pollutants that accumulate in bottom sediments, are commonly used as bioindicators in aquatic habitats [21,22,23]. Furthermore, zoobenthos are an important component of the trophic pyramid of lakes and rivers, providing food for many water organisms (fish and birds, among others) [1,24]. Hence, in the aquatic environment, benthic organisms constitute an important link between primary producers and secondary consumers, play an important part in the trophic cycle, and promote the decomposition of organic matter through the consumption and breakdown of plant and animal tissues [24,25].

Although rivers are habitats rich in benthic macroinvertebrates [22,26], their assemblages depend on many environmental factors, mainly hydrological and physicochemical. The habitats of benthic macroinvertebrates are shaped by water currents that, in turn, affect physicochemical conditions [27]. It has been repeatedly shown that increasing the trophic level of a watercourse has an adverse effect on the river’s invertebrate communities. Rapid impoverishment of such communities is observed, for example, under the effect of pollution of montane watercourses, especially with increased levels of nitrogen compounds [28]. This occurs because many of the component taxa are sensitive to changes in the concentration of biogenic substances [19,29].

Likewise, an increase in water temperature and a decrease in oxygen concentration may have a negative effect on the number of macrozoobenthos taxa in rivers [26,30]. Bottom type is another important variable that shapes distribution of macrozoobenthos. As shown by Gonzáles and Graça [31], watercourses with large quantities of detritus harbour fewer macrozoobenthos taxa, resulting in lower biodiversity. A great deal of research on the effect of environmental variables on macroinvertebrate assemblages has been conducted in large rivers [10,22,26,32] and in small watercourses of montane character [11,33]. However, there is little information on the crucial environmental and hydrological factors that affect the density and diversity of the macrozoobenthos communities of small, heavily eutrophicated watercourses of the European Central Plains ecoregion [34].

It appears that in small, relatively shallow watercourses, the effect of environmental variables on macrozoobenthos assemblages may be more pronounced than in large rivers because of their considerable hydrological variation [27]. Thus, knowledge of the effect of selected biotic and abiotic factors is essential for the protection of these valuable communities, which are important components of trophic pyramids in small rivers.

The objectives of this study were to (i) assess the environmental conditions in small watercourses of the two study areas; (ii) compare the abundance, richness, diversity, and taxonomic structure of the macrozoobenthos assemblages; and (iii) identify which factors have the greatest effect on diversity and density of this group of organisms in small watercourses located in agricultural areas of the ecoregion of European Central Plains.

## 2. Materials and Methods

### 2.1. Study Area

Two areas within the ecoregion of European Central Plains were selected for study, with 10 small watercourses; these sites are indicated by blue circles in Figure 1: Płonia (Pl), Myśla (My), Tywa (Tyw), Rurzyca (Rur), and Wardynka (War) in northwestern Poland (NW PL); and Kanał Habdziński (KH), Zielona (Ziel), Czarna Cedron (CC), Kraska (Kr), and Molnica (Ml) in central Poland (CE PL).

The watercourses were mostly located in agricultural catchment areas; only the Wardynka catchment region was dominated by forests. Table 1 presents the characteristics of the study sites. The environmental conditions were determined based on Corine Land Cover 2018 [35], and the catchment land use was calculated with the QGIS 3.18 software. The agricultural areas included arable land, perennial crops, meadows, pastures, and fruit orchards. The semi-natural areas were forests, semi-natural ecosystems, and scrubland.

### 2.2. Sampling

Macrozoobenthos samples were collected in October of the years 2017–2020. They were taken with a hand dredge on a handle of mesh size 500 µm and entrance width 25 cm. The dredge was pulled along a section of the river, directly against the current, which disturbed the sediment surface in front of the net over an area of 25 cm × 140 cm, taking the surface layer of sediments to the depth of 5 cm. The surface from which the sample was taken was calculated based on the distance over which the dredge was pulled and the width of dredge entrance. A total of 40 samples were taken (each containing four subsamples from different parts of the watercourse). Each sample, after the manual removal of macrophytes, stones, and other solid components of the sediment, was rinsed three times in a sieve (500 µm); all the collected animals were immediately fixed in 4% formalin in the field and then transferred to 70% ethyl alcohol. In the laboratory, the material was identified and counted using a Zeiss Discovery V12 stereomicroscope (Carl Zeiss, Dresden, Germany). In the case of abundant organisms, a subsample method was examined. Specimens of particular taxa whose organisms were large and not highly abundant were removed from the sample. Our subsample method involved random selection of a subset of invertebrates that is used as a representation of the entire sample population [36]. All material was examined in the laboratory using artificial lighting at 10× magnification. The organisms were identified to the lowest possible definable taxonomic rank.

Along with macrozoobenthos sampling, physiochemical measurements were taken directly in the field. Temperature, conductivity (EC), and water oxygenation were measured using a multiparameter sensor HQD30 produced by Hach (Düsseldorf, Germany). At the same time, water samples were taken from the rivers following the current standards [37,38] (PN-EN ISO 5667-6:2016-12, PN-EN ISO 5667-3:2018-08) to determine concentrations of N-NO_3_, N-NH_4_, and P-PO_4_. The concentration of nitrogen forms and phosphates was determined colorimetrically using an automatic flow analyser produced by Skalar (Breda, The Netherlands). Flow velocity was measured using a SENSA RC2 electromagnetic meter coupled with an RV2 probe produced by Quantum Dynamics Ltd. Aqua Data Services Division (Oxfordshire, United Kingdom).

### 2.3. Data Analyses

Shapiro–Wilk tests were performed using R software [39], to test the normality of the distribution for the density of taxa and the physicochemical indices [40]. Significance of differences between the variables was determined using ANOVA followed by a Tukey’s post hoc test [41].

Statistical multidimensional analyses included Canonical Correspondence Analysis (CCA) [42] and Ward clustering [43]. The first four axes selected in the CCA were those that, in combination, explained the greatest part of the total variation.

The significance of correlation between the physicochemical parameters and the density of organisms was tested using the Generalised Linear Model (GLM) [44]. Linear regression was also performed to test the relationships between the total density and the individual physicochemical and hydrological parameters.

## 3. Results

### 3.1. Environmental Characterisation of the Studied Watercourses

The watercourses were characterised by varied physicochemical conditions and stream morphology. Among the studied parameters, the highest coefficient of variation (CV) was recorded for flow (120.5%) and the content of ammonia nitrogen N-NH_4_ (118.7%), while the lowest was shown by pH (2.84%) (Table 2). The high content of biogenes (N-NO_3_, N-NH_4_, and P-PO_4_) and high electrolytic conductivity, especially in the rivers Kraska, Molnica, and Zielona, indicated a high trophic level (Table 2).

With respect to the characteristics of the studied watercourses, cluster analysis yielded three clusters of similar physicochemical and hydrological parameters (Figure 2). In this respect, the Kanał Habdziński showed significant departure from the remaining watercourses.

### 3.2. Macroinvertebrate Diversity and Abundance Patterns

A total of 105 taxa were recorded in the studied sites during the 4 years of study; these were classified into the 17 higher taxonomies. The highest number of taxa was recorded in the river Płonia (13 taxa), the smallest in the Kraska (7) and Molnica (9). The highest total density was observed in the Kanał Habdziński (10015 ind./m^2^), while the lowest was in the rivers Wardynka (512 ind./m^2^) and Zielona (776 ind./m^2^) (Figure 3).

### 3.3. Effects of Environmental Gradients

In the GLM analysis (Table 3), 11 of the 12 studied physicochemical parameters turned out to be statistically significant in the construction of the model for the total density of taxa in the surveyed sites. No relationship was seen between phosphate concentration (P-PO_4_) and total density.

In the CCA analysis (Figure 4a,b), axes 1 and 2 explained ~77.4% (56.2% (axis 1) and 21.2% (axis 2) of the variation at eigenvalues of λ1 = 0.169 and λ2 = 0.064, respectively (Table 4).

Among the physicochemical and hydrological variables (Canonical Correspondence Analysis), width (0.366) and depth (0.250), as well as P-PO_4_ (−0.307), EC (−0.239), and O_2_ (%) (−0.204) showed the strongest correlation with the first axis (Table 5). P-PO_4_ (0.339), EC (0.213), as well as flow (−0.489), depth (−0.345), and width (−0.226) had the strongest correlations with axis 2. Oligochaeta (1.75) and Gastropoda (0.654), as well as Chironomidae (−1.096), Hirudinea (−0.827), and Bivalvia (−0.709) had the strongest correlation with axis 1. Chironomidae (1.44) and Oligochaeta (0.589), as well as Hirudinea (−1.961), Asellidae (−1.236), and Gastropoda (−0.770) had the strongest correlation with axis 2.

Figure 4a shows that higher taxa prefer sites with smaller depth and width, such as Gammaridae, Heteroptera, Simuliidae, and Chironomidae. The position of Simuliidae, Gammaridae, and Heteroptera near the vectors of water oxygenation (in mg/dm^3^ and in %) indicates the importance of this factor. Such taxa as Ephemeroptera, Diptera, Decapoda, Plecoptera, and Coleoptera, are positioned in the bottom left part of the graph, near the temperature and pH vectors, which can be regarded as favouring these two variables as opposed to the N-NH_4_ content. The right bottom part of the graph includes taxa for which hydrological variables are important—depth and width for Gastropoda, and speed and flow for Asellidae and Trichoptera. These taxa are sensitive to the ammonium ion content of water. Oligochaeta, which occur in most sites, were remote from all the vectors of environmental variables and seemed not to be affected by any of the physicochemical and hydrological variables. In Figure 4b, the CCA ordination did not show any clear differences between the surveyed sites.

The factors with the greatest effect on diversity at higher taxon levels (Table 6) were percent oxygen saturation, depth, and flow; the smallest effect was exerted by speed and N-NO_3_. The total density of organisms was affected by flow and O_2_ (mg/dm^3^). Among the physicochemical variables that were important for biodiversity, temperature had no effect on any taxon, due to similar water thermic values. EC affected density of four taxa to a large extent; these were Hirudinea, Asellidae, Chironomidae, and Gastropoda. Oxygen was very important for Hirudinea and Ephemeroptera, but had little effect on Asellidae and Bivalvia. Presence of biogenic N-NO_3_, N-NH_4_, and P-PO_4_ had an effect on Oligochaeta, Hirudinea, Asellidae, Ephemeroptera, and Gastropoda. No effect of physicochemical variables on density was observed for nine taxa (Table 6); this was probably due to the small number of sites of their occurrence affecting the analysis.

## 4. Discussion

The diverse physicochemical and morphological conditions of the watercourses had an effect on the distribution and density of benthic organisms. The total density of macrozoobenthos was, statistically, significantly influenced by 11 of the 12 studied physicochemical and morphological parameters of the watercourses. The greatest effects on the various groups of macrozoobenthos that were exerted included oxygen and nitrate nitrogen content and water temperature. Water quality in the studied rivers and, indirectly, their macrozoobenthos communities may be affected by agricultural land use in their catchment areas (Table 1) [45]. The European Central Plains ecoregion is dominated by agriculture [46] and, as a result of intensive fertilisation of fields, there is a concomitant increased in nutrients and pesticides in catchment areas [45]. In addition to the increased deposition of fine-grained sediments and the agricultural pollutants contained within, physicochemical and hydrological conditions are altered from pristine settings [47]. Increased contents of fine sediment often led to an increase in the organic matter content, resulting in a decrease in oxygen content and an increase in the content of biogenic substances [48]. This may be the cause of the observed differences in the environmental conditions among the watercourses we studied. Changes in environmental conditions as a result of land use may lead to a decrease in biodiversity, biomass, and density of sensitive species of aquatic animals, thereby leading to changes in the biotic composition [47,49]. Small watercourse ecosystems are especially sensitive to the anthropopressures noted above [50].

Developmental stages of freshwater species are affected by habitat quality, therefore, it is important that we understand how they are influenced by specific anthropopressures [51]. During our four-year study, we recorded 105 taxa, including Oligochaeta and Hirudinea, Bivalvia and Gastropoda, Crustacea (Decapoda, Gammaridae and Asellidae), Odonata, Plecoptera, Trichoptera, Coleoptera, and Diptera (Chironomidae, Simuliidae). The observed diversity and density of benthic communities are typical of small watercourses in many countries [52,53,54]. It should be emphasised, however, that most species of macrozoobenthos in each of the studied rivers are euryoecious and inhabit waters with a wide range of environmental parameters (i.e., Oligochaeta of the family Enchytraeidae and the crustacean *Asellus aquaticus*). Stenoecious species (e.g., *Ameletus inopinatus* and *Parapoynx stratiotata*) occur in the river Wardynka, whose hydromorphology is close to that of montane streams. The observed differences in the structure of macrozoobenthos assemblages, with a distinct predominance of pollution-tolerant taxa, may result from the relations between the macroinvertebrates and the habitat conditions. Such relations are complex and include the direct and indirect effects of many factors [55] that may be specific to a watercourse or even to a site [52,54].

The most important biotic factors are vegetation [56] and competition between species [57]. Among the abiotic factors that affect macrozoobenthos, the most often mentioned are flow speed [58], bottom substratum [59], water temperature [60], oxygen content [61], and dissolved substances, including biogenes [57,59]. It should be emphasised that the effect of individual environmental parameters on benthic organisms depends on the tolerance, sensitivity, and adaptability of each particular species [20].

It is generally thought that temperature is one of the most important factors affecting both the number of taxa and their density [62]. Temperature has an effect on metabolic rate [63], reproduction [57], emergence patterns of insects, and body size [57]. As shown by Živić et al. [62] and Krepski et al. [26], increased water temperature has a negative effect on the abundance of some groups of macrozoobenthos. However, based on studies in the Follonica Bay (the western part of the Mediterranean Sea), Lardicci et al. [64] claimed that the increase in temperature had no significant effect on the abundance and structure of macrozoobenthos. The effect of water temperature can vary in different groups of zoobenthos depending on the sensitivity of particular taxa. Results of the CCA analysis for the 10 watercourses confirmed the effect of water temperature on the density of Ephemeroptera, Coleoptera, Diptera, Plecoptera, and Decapoda, with the latter affected to the greatest extent. Moreover, observations in natural habitats [65] and in laboratory conditions [66] indicate that the last group is sensitive to increased temperature.

Today, with increase in water temperature due to global warming, there is an accompanying decrease in oxygen concentrations; this often occurs in combination with the appearance of toxic sulphur hydrogen, leading to changes in benthic organism communities [67]. Each group of benthic organisms requires a specific level of dissolved oxygen appropriate to its needs [68]. Some nematode species can survive prolonged anoxia [69] or even live and reproduce in conditions of oxygen deficit [70]. According to Giere [67], macrofauna have greater overall oxygen requirements than meiofauna, and particular species inhabit waters with their specific preferred oxygen conditions. For example, the EPT complex (Ephemeroptera + Plecoptera + Trichoptera) occurs only in pure, well-oxygenated waters and is sensitive to anthropogenic and environmental disturbances [71]. Such waters also harbour an array of species of the family Simuliidae, though some simuliids are more tolerant (e.g., *S. ornatum* and *S. equinum*) [72].

Several studies point to the important role of electrolytic conductivity for macrozoobenthos. However, as demonstrated by Braukmann and Böhme [73], sudden pollution with mineral salts caused a manifold increase in electrolytic conductivity and consequent changes in the taxonomic composition of macrozoobenthos in the river Werra (Germany). Moreover, Piscart et al. [74] observed a decrease in species richness of macrozoobenthos in a small stream caused by an increase in water conductivity, while based on their results from the Czech river Bilina, Orendt et al. [75] showed that conductivity, as the main salinity determinant, is an important factor affecting macrozoobenthos. The importance of conductivity was confirmed in our study of small watercourses. Five groups of benthic organisms (Oligochaeta, Hirudinea, Asellidae, Chironomidae, and Bivalvia) were observed to undergo a significant (confirmed by regression analysis) effect of this parameter on the density. It should be borne in mind that the watercourses also contain organic pollutants, termed organic pollution determinants [75], which, through concentrations of N-NO_3_, N-NH_4_, and P-PO_4_, affect Oligochaeta Hirudinea, Asellidae, Ephemeroptera, Gastropoda, and Bivalvia, as indicated by our regression analysis.

Another important factor affecting invertebrate assemblages is pH. Among other things, it controls switches between harmless nitrogenous ions, such as NH_3_, and toxic forms NH_4_^+^ [76]. The threshold concentration of total nitrogen, which has a negative effect on aquatic organisms, is 3.48 mgN/L at pH 6.5, and 0.25 mgN/L at pH 9.0 [76]. A negative effect of pH on, among other taxa (e.g., Gammaridae and Chironomidae) was indicated by Krepski et al. [26] and Golovatyuk et al. [77]. Regression analysis indicates that pH has a very strong effect on Oligochaeta and a weak effect on Bivalvia, while the CCA analysis showed a direct effect of pH on Trichoptera, Decapoda, and Hirudinea. Precise experimental data were presented by Berezina [78], who specified ranges of pH tolerance and optimum values for many taxa. For example, *Asellus aquaticus*, according to Berezina [78], tolerates pH within the range of 4.5–11.0, and the optimum is 6.5–9.0.

Flow is a characteristic feature of lotic waters, shaping the environmental conditions for macrozoobenthos [79]. Several authors found that flow, along with temperature, was the most important factor affecting macrozoobenthos structure [30]. This is compatible with our results, which suggest flow and depth were the only hydrological parameters that affected density of macrozoobenthos, and flow had a significant effect on Hirudinea. The magnitude of flow shapes the habitat conditions of macrozoobenthos in watercourses through its direct influence on the bottom substratum and vegetation and its indirect influence on the physicochemical conditions of the water [30]. A sudden increase in flow may cause washing out and downstream drift of many species of benthic organisms [80]; canalised watercourses host macrozoobenthos assemblages that have poor diversity [81].

Due to the numerous interactions of the physicochemical and hydrological factors, small watercourses are difficult to study [50,56,57]. Benthophagous fishes occur in the watercourses we studied and, even when in low density, may influence occurrence of benthic organisms [82]. Fish abundance in the surveyed sites was low (ranging from 0.00 to 0.26 fish/m^2^ of water surface area), but, as Nicola et al. [83] noted, physicochemical conditions have a greater effect on the zoobenthos community than on the presence of fish.

Our results indicate that eutrophicated watercourses under strong anthropopressure in agricultural areas are characterised by low macrozoobenthos species richness and diversity, and that physicochemical and hydrological conditions have a great effect on the density of these organisms.

## 5. Conclusions

The small watercourses we studied had a reduced oxygen content and a high level of nitrogen and phosphorus compounds, which was related to their location in agricultural areas. Among the physicochemical parameters, the greatest effect on density and abundance of macrozoobenthos taxa was exerted by oxygen concentration and nitrate nitrogen content. Changes in oxygen are very often related to changes in water temperature. It was found that, in all the studied watercourses, the temperature had a significant effect on the density of macrozoobenthos, especially Decapoda, Ephemeroptera, Coleoptera, Diptera, and Plecoptera. Among morphological factors, an important effect on species diversity of macrozoobenthos in small watercourses was exerted by riverbed width and depth. The highest number of taxa and the highest density were recorded in the rivers Płonia and Kanał Habdziński, which were characterised by large width and depth.

We posit that to increase species richness of benthic macroinvertebrate in streams located in agricultural catchments, runoff of nutrients should be limited by having buffer strips adjacent to the cultivated fields planted with shade trees. The trees will have the added effect of shading the water course, thereby reducing elevated temperatures. This practice will improve physicochemical conditions of watercourses. Institutions responsible for protection of the aquatic environment should systematically monitor the quality of these waters and determine the load of nutrients brought in from cultivated fields. This information will be valuable in predicting changes in the aquatic environment and can be used to implement abatement practices to limit habitat degradation before it becomes serious.

In addition, further research is needed on all benthic macroinvertebrate in small watercourses to better understand their sensitivity and resistance to hydrochemical conditions and changes in habitat conditions, while taking into account hydromorphological features of these watercourses. This will facilitate the understanding of the impact relationships and anthropogenic changes in water ecosystems have on individual macrozoobenthos species. It will also support decisions to introduce restrictions to human activity in agricultural areas.

## Figures and Tables

**Figure 1 animals-12-00606-f001:**
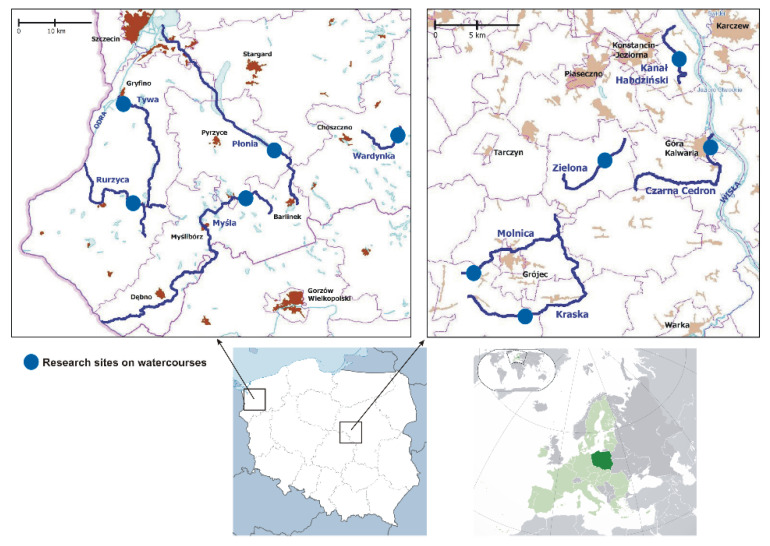
Study areas within the ecoregion of the European Central Plains. **Lower right** panel, Poland (dark green); **lower left** panel, location of two study areas with Poland; **upper** panels, locations of the study areas (blue circles).

**Figure 2 animals-12-00606-f002:**
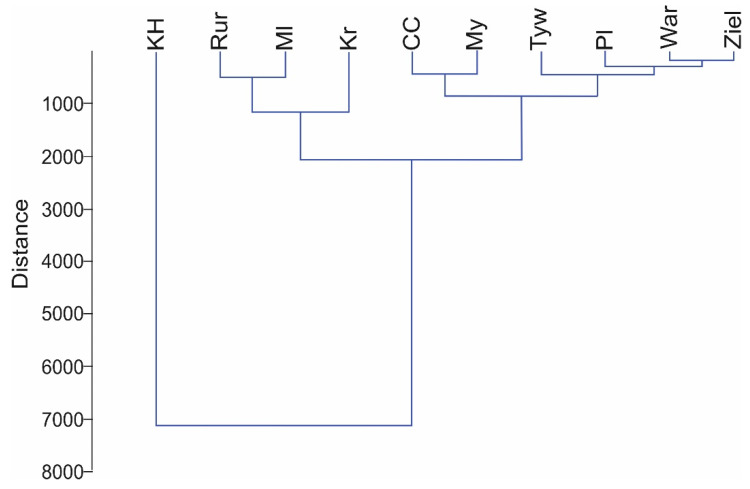
Results of cluster analysis for the studied watercourses based on environmental parameters. Abbreviations: Płonia (Pl), Myśla (My), Tywa (Tyw), Rurzyca (Rur), Wardynka (War), Kanał Habdziński (KH), Zielona (Ziel), Czarna Cedron (CC), Kraska (Kr), and Molnica (Ml).

**Figure 3 animals-12-00606-f003:**
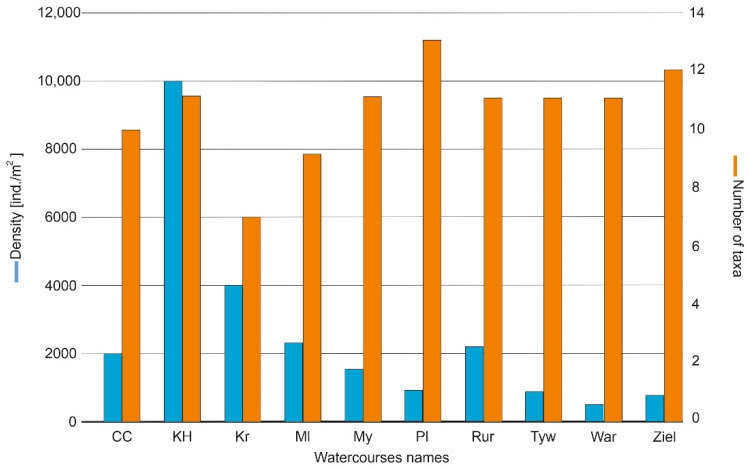
Number of taxa and density (ind./m^2^) of the surveyed sites.

**Figure 4 animals-12-00606-f004:**
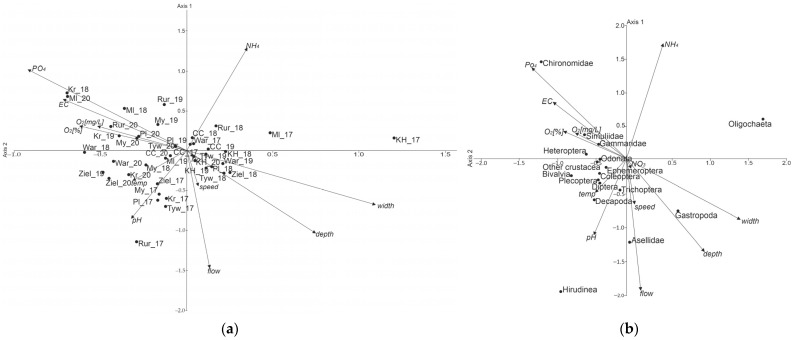
CCA graph for the surveyed sites: (**a**) sites, (**b**) taxa.

**Table 1 animals-12-00606-t001:** Characteristics of sampling sites in the 10 watercourses.

Watercourse Name	Region	Geographical Coordinates	Surface Area of Catchment [km^2^]	Land Use and Characteristics of Catchment ^1^
Płonia	NW PL	N 53.132223 E 15.133033	174.59	A—54%, FO—30%, M—8%, U—5%, W—2%, MA—1%
Myśla	NW PL	N 52.999399 E 14.977928	143.16	A—68%, FO—22%, M—7%, U—1%, W—1%, MA—1%
Tywa	NW PL	N 53.226618 E 14.488537	270.23	A—57%, FO—28%, M—7%, U—4%, W—3%, MA—1%
Rurzyca	NW PL	N 52.976910 E 14.543279	83.03	A—58%, FO—24%, M—12%, U—3%, MA—2%, W—1%
Wardynka	NW PL	N 53.160318 E 15.621905	25.16	FO—51%, A—36%, M—13%
Kanał Habdziński	CE PL	N 52.1099139 E 21.1598859	21.57	A—56%, M—16%, U—16%, FO—7%, MA—5%
Zielona	CE PL	N 51.9721621 E 21.0445781	26.72	A—59%, M—19%, FO—13%, U—7%, O—2%
Czarna Cedron	CE PL	N 51.9738150 E 21.2179701	69.48	O—34%, FO—32%, A—18%, U—13%, M—3%
Kraska	CE PL	N 51.8039040 E 20.8796420	27.14	O—44%, A—30%, FO—14%, M—8%, U—4%
Molnica	CE PL	N 51.8548140 E 20.8109830	13.25	O—68%, A—19%, FO—13%

^1^ NW PL—northwestern Poland, CE PL—central Poland; A—arable land, M—meadows, O—orchards, FO—forest, U—urban areas, W—water (reservoirs, rivers), MA—marshes.

**Table 2 animals-12-00606-t002:** Physicochemical and hydrological conditions of the watercourses with results of Tukey’s test.

Feature	CC	KH	Kr	Ml	My	Pl	Rur	Tyw	War	Ziel	Average	SD	CV
speed	0.06 ^a^	0.09 ^a^	0.09 ^a^	0.07 ^a^	0.32 ^ab^	0.19 ^a^	0.31 ^ab^	0.60 ^b^	0.84 ^b^	0.13 ^a^	0.27	0.26	96.20
flow	0.16 ^b^	0.27 ^c^	0.03 ^a^	0.01 ^a^	0.12 ^b^	1.53 ^d^	0.79 ^d^	1.09 ^d^	0.30 ^c^	0.05 ^a^	0.43	0.52	120.45
width	5.62 ^a^	6.20 ^d^	1.87 ^ab^	0.99 ^a^	2.12 ^b^	8.96 ^e^	4.55 ^c^	4.53 ^c^	2.81 ^b^	2.62 ^b^	4.03	2.43	60.34
depth	0.55 ^c^	0.49 ^c^	0.55 ^c^	0.49 ^c^	0.24 ^b^	0.08 ^a^	0.13 ^a^	0.76 ^c^	0.44 ^b^	0.35 ^b^	0.33	0.23	70.02
temp	18.48 ^a^	17.40 ^a^	18.48 ^a^	17.40 ^a^	16.88 ^a^	17.97 ^a^	16.42 ^a^	16.52 ^a^	16.07 ^a^	16.53 ^a^	16.89	1.14	6.76
pH	7.34 ^a^	7.21 ^a^	7.34 ^a^	7.21 ^a^	7.74 ^c^	7.51 ^a^	7.49 ^a^	7.76 ^c^	7.38 ^a^	7.86 ^b^	7.54	0.21	2.84
EC	501 ^a^	586 ^a^	501 ^a^	586 ^a^	827 ^c^	819 ^c^	774 ^b^	601 ^a^	700 ^b^	592 ^a^	656	114.90	17.50
O_2_ [mg]	5.69 ^b^	5.29 ^b^	5.69 ^b^	5.29 ^b^	6.82 ^c^	5.58 ^bc^	5.23 ^b^	5.91 ^c^	4.01 ^a^	6.87 ^c^	5.95	1.06	17.83
O_2_ [%]	61.58 ^b^	54.85 ^b^	61.58 ^b^	54.85 ^b^	70.41 ^d^	59.78 ^b^	52.85 ^b^	61.24 ^c^	39.85 ^a^	69.00 ^c^	61.42	10.68	17.39
N-NO_3_	3.21 ^a^	3.85 ^a^	3.21 ^a^	3.85 ^a^	9.50 ^d^	10.25 ^d^	5.02 ^b^	3.04 ^a^	3.89 ^a^	3.59 ^a^	5.25	2.73	51.93
N-NH_4_	0.10 ^a^	0.35 ^a^	0.10 ^a^	0.35 ^a^	0.16 ^a^	0.18 ^a^	0.53 ^a^	0.27 ^a^	1.56 ^b^	0.30 ^a^	0.37	0.44	118.69
P-PO_4_	0.70 ^a^	0.66 ^a^	0.70 ^a^	0.66 ^a^	0.71 ^a^	1.44 ^a^	0.99 ^a^	1.46 ^a^	1.48 ^a^	0.90 ^a^	0.99	0.34	34.84

Values of parameters in the same row with different indices differ significantly (Tukey’s test, *p* < 0.05), average—mean, SD—standard deviation, CV—coefficient of variation. Abbreviations: Płonia (Pl), Myśla (My), Tywa (Tyw), Rurzyca (Rur), Wardynka (War), Kanał Habdziński (KH), Zielona (Ziel), Czarna Cedron (CC), Kraska (Kr), and Molnica (Ml).

**Table 3 animals-12-00606-t003:** Results of the GLM model for total density of taxa in the surveyed sites.

	Estimate	Std. Error	z Value	Pr (>|z|)
depth	41.926	1.169	35.836	<2.0 × 10^−16^ ***
EC	0.028	0.001	26.481	<2.0 × 10^−16^ ***
flow	−14.356	0.246	−58.318	<2.0 × 10^−16^ ***
N-NH_4_	2.894	0.183	15.835	<2.0 × 10^−16^ ***
N-NO_3_	0.158	0.020	7.711	1.25 × 10^−14^ ***
O_2_ [mg]	24.664	0.509	48.398	<2.0 × 10^−16^ ***
O_2_ [%]	−2.467	0.047	−52.372	<2.0 × 10^−16^ ***
pH	−9.919	0.474	−20.941	<2.0 × 10^−16^ ***
P-PO_4_	0.214	0.204	1.049	0.294
speed	1.073	0.377	2.847	0.004 **
temp.	4.552	0.075	60.404	<2.0 × 10^−16^ ***
width	−1.289	0.108	−11.908	<2.0 × 10^−16^ ***

Significance codes: <0.001—***, 0.001—**

**Table 4 animals-12-00606-t004:** Eigenvalues, % of explained variation and *p* value estimates for the first four of the 12 axes shown by CCA.

Axis	Eigenval	%	*p*
1	0.169	56.21	0.058
2	0.064	21.15	0.172
3	0.027	9.067	0.335
4	0.023	7.45	0.043

**Table 5 animals-12-00606-t005:** Physicochemical factors and higher taxa in the CCA analysis. The data with underline shows important variable for CCA.

	Axis 1	Axis 2	Axis 3	Axis 4
flow	0.044	−0.489	0.134	0.086
width	0.366	−0.226	−0.040	−0.165
depth	0.250	−0.345	0.168	−0.238
EC	−0.239	0.213	0.280	0.025
O_2_ (%)	−0.204	0.101	−0.052	0.069
P-PO_4_	−0.307	0.339	0.147	0.128
Oligochaeta	1.750	0.589	0.261	0.542
Hirudinea	−0.827	−1.961	1.112	1.564
Asellidae	0.029	−1.236	0.198	0.386
Chironomidae	−1.096	1.444	0.937	0.146
Gastropoda	0.654	−0.770	0.505	−2.157
Bivalvia	−0.709	−0.251	0.246	−1.748

**Table 6 animals-12-00606-t006:** Regression. Only statistically significant correlations are shown.

	Olg	Hir	Ase	Eph	Chi	Gas	Biv	No. Taxa	Density
speed							0.024 **	0.021 **	
flow		0.003 ***	0.095 *		0.0538 *			0.008 ***	0.019 **
width	0.041 **		0.0103 **	0.051 *	0.081*		0.037 **		
depth			0.019 **		0.039 **	0.006 ***		0.006 ***	0.077 *
temp									
pH	0.009 ***						0.063 *		
EC	0.079 *	0.018 **	0.012 **		0.031 **	0.029 **		0.054 *	
O_2_ (mg)		0.033 **	0.094 *	0.023 **			0.062 *		0.047 **
O_2_ (%)		0.023 **		0.035 **				0.000 ****	
N-NO_3_			0.048 **				0.073 *	0.043 **	
N-NH_4_		0.028 **		0.016 **					
P-PO_4_	0.036 **	0.078 *	0.007 ***			0.039 **			

Significance codes: <0.001—****; 0.001—***; 0.01—**; 0.05—*. Abbreviations: Olg—Oligochaeta; Hir—Hirudinea; Ase—Asellidae; Eph—Ephemeroptera; Chi—Chironomidae; Gas—Gastropoda; Biv—Bivalvia.

## Data Availability

Not applicable.

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
