# Peer review of "Characterisation of Benthic Macroinvertebrate Communities in Small Watercourses of the European Central Plains Ecoregion and the Effect of Different Environmental Factors"

_animals, 2022, doi:10.3390/ani12050606_

Round 1

Reviewer 1 Report

The manuscript of Adam Brysiewicz and colleagues focus on an interesting environmental topic, focusing on features never treated by other Authors in this study area. I found the study well conducted and the manuscript well written and organized, with some good analysis and results. I have only some minor revision to suggest for improve this manuscript before its publication, that I summarize as follow.

Material and Methods

2.2 Sampling: why the Authors have chosen to sample in October? As it is a survey based on sampling per year for several years, the choice of that single sampling moment should be better argued from a functional point of view.

Results

3.1 Evironmental characteristics, should be renamed Environmental characterization of studied watercourses.

"Figure 2. Results of cluster analysis for the studied watercourses." This figure and the relative caption should be better related to environmental parameters (e.g..studied watercourses features or environmental parameters).

Figure 4 reported in lines 230-231 is erroneously named Figure 2, also in main text the reference to this Figure need to be reviewed.

Conclusion

Conclusion section appears brief and synthetic in the present form. Some result regards oxygen concentration in the water and content of nitrate nitrogen were only cited but not exposed. Please provide a more articulate conclusion of your manuscript, that could give more soundness to your key results.

Best regards

The Reviewer

Author Response

Answer for a Reviewer 1

 Thank you very much for your review. We revised the manuscript as directed and applied all the changes.

 2.2 Sampling: why the Authors have chosen to sample in October? As it is a survey based on sampling per year for several years, the choice of that single sampling moment should be better argued from a functional point of view.

Answer:

October is the first full calendar month of autumn and is the period when the larvae of insects hatched from eggs during the summer period during intensive feeding have reached the second stage of larvae, and sometimes, at high temperatures, it may be the third stage, which is already reaching dimensions that can be obtained. easily selected from benthic trials. If the benthic samples were taken later, some animals, such as leeches Autumn sampling is also recommended by Ntislidou et al. 2021 referring also to the EU Framework Water Directive (FWD). This author points out that there are no differences between the spring and fall trials, and Hill et al. 2016 indicate greater biodiversity in ponds in the autumn period. Similarly, Sporka et al (2006), referring to Hynesa (1972), indicated that autumn is a time of larvae growth of many species of aquatic insects and greater biodiversity in streams.

  • Ntislidou, C.; Bobori, D.; Lazaridou, M. Suggested Sampling Methodology for Lake Benthic Macroinvertebrates under the Requirements of the European Water Framework Directive. Water 2021, 13, 1353. https://doi.org/10.3390/w13101353
  • European Commission. Directive 2000/60/EC of the European Parliament and of the Council of 23 October 2000; Official Journal of European Community: Luxembourg, 2000; pp. 1–72
  • Hill M.J., Sayer C.D., Wood P.J. 2016. When is the best time to sample aquatic macroinvertebrates in ponds for biodiversity assessment? Environ Monit Assess (2016) 188: 194 DOI 10.1007/s10661-016-5178-6
  • Sporka F., Vlek H.E., Bulankova E., Krno I. 2006. Influence of seasonal variation on bioassessment of streams using macroinvertebrates. Hydrobiologia. 566: 543-555
  • Wallace I.D., Wallace B., Philipson G.N. 2003 Keys to the case-bearing caddis larvae of Britain and Irland. FBA 61. Pp. 259

Results

3.1 Evironmental characteristics, should be renamed Environmental characterization of studied watercourses. - done

"Figure 2. Results of cluster analysis for the studied watercourses." This figure and the relative caption should be better related to environmental parameters (e.g..studied watercourses features or environmental parameters). Name changed, added: based on environmental parameters

Figure 4 reported in lines 230-231 is erroneously named Figure 2, also in main text the reference to this Figure need to be reviewed.  – done – we changed number of from Figure 2 to Figure 4

Conclusion

Conclusion section appears brief and synthetic in the present form. Some result regards oxygen concentration in the water and content of nitrate nitrogen were only cited but not exposed. Please provide a more articulate conclusion of your manuscript, that could give more soundness to your key results. – done we changed euryoecious on eurytopic (better characteristics species on wide environmental spectrum). At the end of the Conclusions, the text was supplemented with a sentence summarizing our research.

Reviewer 2 Report

Review

Paper title: Benthic macroinvertebrate communities from small watercourses of the ecoregion of European Central Plains in relation to different environmental factors.

The authors conducted a long-term field study to reveal the most important abiotic factors affecting benthic macroinvertebrate communities in small rivers of Poland. The results from GLM, cluster analysis, and CCA indicated that almost all physico-chemical parameters except for phosphorous had significant effects on the community. The authors concluded that the oxygen concentration, width and depth were the most important factors. This study expands our knowledge regarding environmental variables driving the diversity and abudance of benthic communities in small watercourses.

All these reasons explain the relevance of the paper by Adam Brysiewicz and co-authors submitted to "Animals".

General scores.

The data presented by the authors are original and significant. The study is correctly designed and the authors used appropriate methods. In general, the statistical analyses are performed with good technical standards. We authors conducted careful work which may attract the attention of a wide range of specialists focused on freshwater ecology.

Recommendations.

The authors should include the information about geo-referencing of this study in Abstract.

The authors should use another, more standard, term instead of “macrobatics”

Figure 2 caption. The authors should define each abbreviation.

Figure 3. The authors used a line for the number of taxa that is incorrect in this case.

The authors should use “ind./m2” or “ind. m–2” instead of  “indiv./m2

The authors discussed the levels of Simpson index (L 282-286) and Evenness (L 388-390) but these results are not presented in the corresponding section. The authors should update the text.

Specific comments.

L 48. Change “adaptable” to “and adaptable”

L 60. Change “aquatic  environment” to “the aquatic  environment”

L 63. Change “zoobenthos  is” to “zoobenthos  are”

L 65. Change “aquatic  environment, it  constitutes” to “the aquatic  environment, benthic organisms constitute”

L 66. Change “plays an important part in the trophic cycle and favours” to “play an important part in the trophic cycle and favour”

L 75. Change “level” to “levels”

L 78. Change “increase in water temperature and decrease in oxygenation have” to “an increase in water temperature and a decrease in oxygenation may have”

L 121. Change “October  in” to “October”

L 132. Change “stereomicroscope Zeiss Discovery V12 (Carl Zeiss, Germany)” to “a Zeiss Discovery V12 stereomicroscope (Carl Zeiss, Germany)”

L 133. Change “subsample method” to “the subsample method”

L 135. Change “consisted in random selection of a subset of invertebrates and using it as” to “is a random selection of a subset of invertebrates and used as a”

L 139. Change “Parallel  to” to “Along with”

L 140. Change “measured with” to “measured with a”

L 143. Change “concentration” to “concentrations”

L 145. Change “with automatic” to “with an automatic”

L 146. Change “with  electromagnetic” to “with  an electromagnetic”

L 151. Change “normalcy” to “normality”

L 153-154. Change “ANOVA, and Tukey test was applied as post hoc test [41].” to “ANOVA followed by a Tukey’s post hoc test [41].”

L 155. Change “CCA” to “Canonical Correspondence Analysis (CCA)”

L 157. Change “variation” to “the total variation”

L 160. Change “dependence” to “relationships”

L 166. Change “content” to “the content”

L 167. Change “biogenes” to “biogenics”

L 172. Change “Tukey” to “Tukey’s”

L 189. Change “density” to “density (ind./m2)”

L 199. Change “variation” to “the variation”

L 203. Delete this footnote.

L 205. Change “Correlation” to “Correspondence”

L 208. Change “were” to “had”

L 210. Change “the  best  correlated” to “best  correlated”

L 212. Change “the  best  correlated” to “best  correlated”

L 220. Change “indicate” to “indicates”

L 238. Change “little effect” to “a little effect”

L 246. Delete “test”

L 299. Change “biogenes” to “biogenics”

L 301. Change “concentration of O2  [%] and N-NO3 was” to “the concentrations of O2  [%] and N-NO3 were”

L 307. Change “to affect” to “affecting”

L 314. Change “changes  on” to “can vary in”

L 315. Change “depends” to “depending”

L 315. Change “CCA” to “the CCA”

L 321. Change “the  global  warming  and  increase” to “global  warming  and an  increase”

L 334. Change “demonstrated” to “demonstrated by”

L 355. Change “CCA” to “the CCA”

L 377. Change “much” to “the majority of”

L 392. Change “Most frequent” to “The most frequent”

L 393. Change “wide  spectrum” to “a wide  spectrum”

L 443. “Corbicula fluminea” should be italicized.

Author Response

Responses for the Reviewer 2

 Thank you very much for your review. We revised the manuscript as directed and applied all the changes.

Recommendations.

The authors should include the information about geo-referencing of this study in Abstract. - we have included data on the mean coordinates of research areas

The authors should use another, more standard, term instead of “macrobatics” –done (change on macroinvertebrates species, and aquatic animals)

Figure 2 caption. The authors should define each abbreviation.– done – we put in abbreviation in Table 2 and in Figure 2.

Figure 3. The authors used a line for the number of taxa that is incorrect in this case.

The authors should use “ind./m2” or “ind. m–2” instead of  “indiv./m2”– done – we changed indiv./m2 on ind./m2–done – we hanged plot type from line to bar plot for number of species and number for specimen, we added captions under the lines

The authors discussed the levels of Simpson index (L 282-286) and Evenness (L 388-390) but these results are not presented in the corresponding section. The authors should update the text. Done – (We deleted part about Simpson’s and Evenness index from abstract and discussion) .

Specific comments.

L 48. Change “adaptable” to “and adaptable”done

L 60. Change “aquatic  environment” to “the aquatic  environment”done

L 63. Change “zoobenthos  is” to “zoobenthos  are” done

L 65. Change “aquatic  environment, it  constitutes” to “the aquatic  environment, benthic organisms constitute” done

L 66. Change “plays an important part in the trophic cycle and favours” to “play an important part in the trophic cycle and favour” done

L 75. Change “level” to “levels”done

L 78. Change “increase in water temperature and decrease in oxygenation have” to “an increase in water temperature and a decrease in oxygenation may have”done

L 121. Change “October  in” to “October” done

L 132. Change “stereomicroscope Zeiss Discovery V12 (Carl Zeiss, Germany)” to “a Zeiss Discovery V12 stereomicroscope (Carl Zeiss, Germany)”done

L 133. Change “subsample method” to “the subsample method” done

L 135. Change “consisted in random selection of a subset of invertebrates and using it as” to “is a random selection of a subset of invertebrates and used as a” done

L 139. Change “Parallel  to” to “Along with” done

L 140. Change “measured with” to “measured with a” done

L 143. Change “concentration” to “concentrations” done

L 145. Change “with automatic” to “with an automatic” done

L 146. Change “with  electromagnetic” to “with  an electromagnetic”done

L 151. Change “normalcy” to “normality”done

L 153-154. Change “ANOVA, and Tukey test was applied as post hoc test [41].” to “ANOVA followed by a Tukey’s post hoc test [41].”done

L 155. Change “CCA” to “Canonical Correspondence Analysis (CCA)”done

L 157. Change “variation” to “the total variation”done

L 160. Change “dependence” to “relationships”done

L 166. Change “content” to “the content”done

L 167. Change “biogenes” to “biogenics” we didn't change it because there is probably a misunderstanding, because biogenes are NO3, NO2, NH4, PO4, but biogenics are combine life and machine

L 172. Change “Tukey” to “Tukey’s” done

L 189. Change “density” to “density (ind./m2)”done

L 199. Change “variation” to “the variation”done

L 203. Delete this footnote.done

L 205. Change “Correlation” to “Correspondence” done

L 208. Change “were” to “had”done

L 210. Change “the  best  correlated” to “best  correlated”done

L 212. Change “the  best  correlated” to “best  correlated”done

L 220. Change “indicate” to “indicates”done

L 238. Change “little effect” to “a little effect”done

L 246. Delete “test”done

L 299. Change “biogenes” to “biogenics”, we didn't change it because there is probably a misunderstanding, because biogenes are NO3, NO2, NH4, PO4, but biogenics are combine life and machine

L 301. Change “concentration of O2  [%] and N-NO3 was” to “the concentrations of O2  [%] and N-NO3 were” done

L 307. Change “to affect” to “affecting”done

L 314. Change “changes  on” to “can vary in”done

L 315. Change “depends” to “depending”done

L 315. Change “CCA” to “the CCA”done

L 321. Change “the  global  warming  and  increase” to “global  warming  and an  increase”done

L 334. Change “demonstrated” to “demonstrated by”done

L 355. Change “CCA” to “the CCA” done

L 377. Change “much” to “the majority of”done

L 392. Change “Most frequent” to “The most frequent” done

L 393. Change “wide  spectrum” to “a wide  spectrum”done

L 443. “Corbicula fluminea” should be italicized. done
